# The LINC Complex Inhibits Excessive Chromatin Repression

**DOI:** 10.3390/cells12060932

**Published:** 2023-03-18

**Authors:** Daria Amiad Pavlov, CP Unnikannan, Dana Lorber, Gaurav Bajpai, Tsviya Olender, Elizabeth Stoops, Adriana Reuveny, Samuel Safran, Talila Volk

**Affiliations:** 1Department of Molecular Genetics, Weizmann Institute of Science, Rehovot 7610001, Israel; 2Bayer AG, 51368 Leverkusen, Germany; 3Department of Chemical and Biological Physics, Weizmann Institute of Science, Rehovot 7610001, Israel

**Keywords:** LINC complex, muscle, nuclear mechanobiology, epigenetics, chromatin repression

## Abstract

The Linker of Nucleoskeleton and Cytoskeleton (LINC) complex transduces nuclear mechanical inputs suggested to control chromatin organization and gene expression; however, the underlying mechanism is currently unclear. We show here that the LINC complex is needed to minimize chromatin repression in muscle tissue, where the nuclei are exposed to significant mechanical inputs during muscle contraction. To this end, the genomic binding profiles of Polycomb, Heterochromatin Protein1 (HP1a) repressors, and of RNA-Pol II were studied in *Drosophila* larval muscles lacking functional LINC complex. A significant increase in the binding of Polycomb and parallel reduction of RNA-Pol-II binding to a set of muscle genes was observed. Consistently, enhanced tri-methylated H3K9 and H3K27 repressive modifications and reduced chromatin activation by H3K9 acetylation were found. Furthermore, larger tri-methylated H3K27me3 repressive clusters, and chromatin redistribution from the nuclear periphery towards nuclear center, were detected in live LINC mutant larval muscles. Computer simulation indicated that the observed dissociation of the chromatin from the nuclear envelope promotes growth of tri-methylated H3K27 repressive clusters. Thus, we suggest that by promoting chromatin–nuclear envelope binding, the LINC complex restricts the size of repressive H3K27 tri-methylated clusters, thereby limiting the binding of Polycomb transcription repressor, directing robust transcription in muscle fibers.

## 1. Introduction

The Linker of Nucleoskeleton and Cytoskeleton (LINC) complex, implicated in mechanical coupling between the cytoplasm and nucleoplasm, has been proposed to control chromatin organization and the epigenetic state of chromatin [1,2,3,4]. This complex consists of Nesprin proteins whose membrane-KASH domain is inserted into the outer nuclear membrane; their N-terminus associates with various cytoskeletal components. In the perinuclear space, Nesprin tetramers bind covalently to SUN domain tetramers, which further associate with the nuclear lamina and with chromatin [5,6,7,8,9]. The functional contribution of the LINC complex for human health is displayed by numerous diseases associated with mutations in genes coding for components of the LINC complex, including Emery Dreifuss Muscular Dystrophy (EDMD), Arthrogryposis, Cerebral ataxia, Dilated Cardiomyopathy (DCM) and others [10,11,12,13]. Whereas the primary cause of LINC-associated diseases is not entirely clear, lack of functional LINC complex has been recently suggested to alter chromatin organization resulting in defects with gene transcription and aberrant tissue function. For example, depletion of Nesprin-3 leads to nucleus collapse and loss of genome organization in adult rat cardiomyocytes [14], LINC disruption in embryonic cardiomyocytes led to global chromatin and H3K9me3 rearrangement [15] and knockout of Nesprin 2 in fibroblasts led to dissociation of HP1b-associated chromatin from the nuclear envelope and an overall reduction in H3K9me3 levels [16]. Meanwhile, plants deficient in components of the LINC complex, such as KASH (wifi) and SUN (sun1 sun4 sun5 triple mutant), show altered nuclear shape, increased distance of chromocenters from the nuclear periphery, altered heterochromatin organization and reactivation of transcriptionally silent repetitive sequences [3]. Likewise, mouse keratinocytes lacking SUN proteins exhibited precocious epidermal differentiation and loss of repressive chromatin H3K27me3 mark on differentiation-specific genes [17]. In oligodendrocytes, silencing of the Nesprin Syne1 gene resulted in aberrant histone marks, chromatin reorganization and impaired gene transcription [18]. Furthermore, in *S. cerevisiae* Mps3, a SUN homologue is involved in the recruitment of heterochromatic sequences such as telomeric repeats to the nuclear envelope (NE), an essential process needed for spindle formation in the course of chromosome segregation [19]. These studies provide strong evidence to the role of the LINC complex in regulation of chromatin epigenetic state, but also point out the developmental stage and tissue-specific mechanistic differences. Moreover, most of these studies were performed either on cells in culture conditions, or on cells that were not fully differentiated. The contribution of the LINC complex to fully differentiated, non-dividing cells, where chromatin landscape has been stabilized, has yet to be elucidated.

Previously, we demonstrated the functional consequences of disrupted LINC complex on fully differentiated larval muscle fibers, affecting proper larval muscle contractions, and nuclear position along the entire muscle fiber [20,21], as well as the synchronization of myonuclear mechanical dynamics [22]. Here, we investigated the role of the LINC complex in chromatin organization and epigenetic state and show that the LINC complex is needed to minimize chromatin repression. In the present study, we analyzed the genomic binding profile of the transcription repressors Polycomb and HP1a, as well as that of RNA-Pol II, and quantified the levels of repressive epigenetic chromatin marks, tri-methylated H3K27 and H3K9, and active H3K9ac chromatin mark, in *Drosophila* larval muscles lacking functional LINC complex. We found that differentiated muscles lacking functional LINC complex exhibited increased epigenetic repression that correlated with enhanced genome binding of Polycomb and HP1a, together with reduced binding of RNA-Pol II to a set of essential muscle genes. The increased chromatin repression observed in the *SUN/koi* mutant muscle nuclei correlated with increased size of tri-methylated H3K27 chromatin clusters, and with partial translocation of chromatin from the periphery to nuclear center quantified from live 3D imaging. Simulations of the 3D distribution of tri-methylated H3K27 sites indicated that reduced binding of chromatin to the nuclear lamina leads to increased repressive cluster size. These results suggest that the enhanced chromatin repression observed in the LINC mutants was induced by weakening of chromatin binding to the nuclear lamina. Altogether our results imply that by coupling the cytoskeleton and the nucleoskeleton, the LINC complex restrains gene repression in mature muscle fibers.

## 2. Materials and Methods

### 2.1. Fly Stocks and Husbandry

The following stocks were used: *koi^84^/Cyo-dfd-eYfp* (FBst0025105) described previously [23], *koi-RNAi* (FBst0040924), delta KASH (*MSP-300^deltaKASH^*; *klar^mCD4^/TM6B* from J.A Fischer, University of Texas, Austin, TX, USA), *tubP-GAL80^ts^/TM2* (FBst0007017), *GAL4-Mef2.R* (FBst0027390), UAS-Spastin (obtained from V. Brodu, Institute Jacques Monod, Paris, France), ubi-H2B-mRFP/CyO;Dr/TM6B [24], UAS-klar-GFP [20], *Tub-Gal80^ts^*; *Mef2-Gal4* (obtained from F. Schnorrer IBDM, Marseille, France), UAS-2E12LI-EGFP(III)/TM6B (live H3K27me3-GFP mintbody obtained from H. Kimura, Tokyo Institute of Technology, Yokohama, Kanagawa 226-8503, Japan), His2Av-GFP (FBst0005941). Fly lines for Targeted Dam-ID (obtained from A. Brand, The Gurdon Institute, University of Cambridge, Cambridge, UK): UAS-LT3-Dm (FBtp0095492), UAS-LT3-Dm-RpII215 (FBtp0095495), UAS-LT3-Dam-Pc, UAS-LT3-Dam-HP1a [25].

All crosses were carried and maintained at 25, 18, or 29 °C and raised on cornmeal agar. Homozygous *SUN/koi* and ΔKASH mutant larvae were selected by non-Cyo-YFP and confirmed with mis-localization/aggregation phenotype of muscle nuclei. Temporal *SUN/koi* knockdown (*koi*-RNAi) and overexpression of spastin was performed using a combination of *Mef2Gal4* and *tubGal80^ts^* drivers, as follows: embryo collection was performed at 25 °C for 12 h, followed by transfer to a permissive temperature of 18 °C. Larvae were transferred to the restrictive temperature of 29 °C for 3 days (*koi*-*RNAi*) or 1 day (spastin) before the late 3rd instar (wandering larvae). *Tub-Gal80^ts^*; *Mef2-Gal4* fly line was used as control for *SUN*/*koi* mutant, *koi*-*RNAi* and spastin over-expression experiments.

### 2.2. Targeted DamID

We drove the expression of Dam-Polycomb (Cbx8, a component of the PRC1 sub-complex), Dam-HP1a and Dam-Pol II (RpII215) and Dam only in control and SUN/koi mutant muscles using Mef2-gal4 driver (a total of 8 genotypes). Temporal control of Dam expression was achieved with a ubiquitously expressed temperature-sensitive gal80. Each genotype contained 3 independent replicates, with 25 larvae per group. Eggs were laid for 6 h and developed in 18 °C until late larval second instar stage and transferred to 29 °C for 10 h to allow muscle-specific Dam expression. Larvae were then dissected and stored at −80 °C until all triplicates were further processed together following the previously described protocol [26]. Briefly, genomic DNA was extracted and digested with methylation-specific DpnI enzymes that cleave at GATC sites. A double-stranded oligonucleotide adaptor was used to ensure directional ligation. The ligation was followed by digestion with DpnII that cuts only unmethylated GATCs. Finally, a PCR primer was used to amplify adaptor-ligated sequences. These amplified sequences were deep sequenced, the results were further analyzed by a bioinformatic pipeline [27] and reads were normalized to filter out nonspecific Dam binding.

Each sample was processed separately by the damid pipeline to generate normalized ratios for each pair of samples; Dam fused to protein of interest versus Dam only https://github.com/owenjm/damidseq_pipeline (accessed on 6 November 2022). Mean occupancy per gene was then calculated using polii script https://github.com/owenjm/polii.gene.call (accessed on 22 September 2020). The reference genome was taken from flybase https://ftp.flybase.net/releases/FB2016_02/dmel_r6.10/ (accessed on 30 March 2016) and GATC sites from the damid pipeline website. After filtering genes with DamID false discover rate (FDR) < 0.05 (per sample), the three replicates from each group were directed to a clustering algorithm based on pairwise correlation.

To determine the genes with statistically significant occupancy in *SUN/koi* mutant, relative to control, a regression on principal component analysis was performed with a z-score for each of the points around the regression line. The cut-off criteria for significantly altered binding to a gene was set to FDR < 0.05, z-score > 1.96 (2-tailed), and GATC sites > 1, corresponding to an approximately 95% confidence interval. The full list of identified genes for each group is provided in the Appendix A. The full DamID data, including the non-significantly altered genes, are openly available in Mendeley Data at DOI: 10.17632/czjj5m7btr.1 https://data.mendeley.com/datasets/czjj5m7btr/1 (accessed on 17 March 2022). Independently, binding profiles (log2 fold change, normalized to Dam only) of specific genes were further visualized for control and *SUN/koi* mutant using the IGV browser [28]. Gene Ontology Enrichment analysis was performed with Metascape, http://metascape.org (accessed on 17 March 2022) [29] using gene lists from the three clusters identified by K-mean clustering, with a user-defined background composed of all genes identified as occupied by DamID (total 7233 genes).

### 2.3. Immunofluorescence and Antibodies

Quantitative immunofluorescence of epigenetics marks was performed on 3rd instar, wandering larvae that were dissected in phosphate-buffered saline (PBS) and fixed, as previously described [21]. Briefly, dissected larva body walls were fixed in Paraformaldehyde (4% from 16% stock of electron microscopy grade; Electron Microscopy Sciences, Hatfield, PA, USA, 15710) for 20 min, washed several times in PBS with 0.1% TritonX-100, and mounted in Shandon Immu-Mount (Thermo Fisher Scientific, Waltham, MA, USA). The following primary antibodies were used: rabbit anti-H3K9ac (Abcam, Cambridge, UK, AB4441), rabbit anti-H3K9me3 (Abcam, AB176916), mouse anti-H3K27me3 (Abcam 6002). The following conjugated secondary antibodies were used: Alexa Fluor 555 goat anti-rabbit (Renium, #A27039) and Alexa Fluor 647 goat anti- mouse (Renium, #A21235). Hoechst 33342 (1 µg/mL; Sigma-Aldrich, St. Louis, MO, USA) was used to label DNA.

### 2.4. Live Imaging

For imaging live nuclei in their intrinsic environment, a minimal constraint device for *Drosophila* larvae was designed in our laboratory, to be placed on top of a confocal microscope stage, as previously described [22]. For stationary 3D live, in vivo imaging of muscle nuclei, a selected wandering 3rd instar larva was immersed in water for ~4 h to decrease its movement (larval movement could be restored by exposure to air). For each larva, at least three nuclei were imaged from randomly chosen muscles along the entire larval body.

### 2.5. Microscopy and Image Acquisition

Immunofluorescence images of epigenetic marks were acquired at 23 °C on a confocal microscope Zeiss LSM 800 with a Zeiss C-Apochromat 40×/1.20 W Korr M27 lens and an Immersol W 2010 immersion medium. The samples were embedded with a coverslip high precision of 1.5 H ± 5 µm (Marienfeld-Superior, Lauda-Königshofen, Germany) and acquired using Zen 2.3 software (blue edition).

Live imaging of H3K27me3-GFP was performed using an inverted Leica SP8 STED3× microscope equipped with internal Hybrid detectors, an acousto-optical tunable filter (Leica Microsystems CMS GmbH, Wetzlar, Germany) and a white light laser excitation laser. Nuclei were imaged with a HC PL APO 86×/1.20 water STED white objective at a scan speed of 400 Hz, a pinhole of 0.8 A.U. and a bit depth of 12. Z-stacks were acquired with 0.308 µm intervals. The acquired images were visualized during experiments using LAS-X software (Leica Application Suite X, Leica Microsystems CMS GmbH, Wetzlar, Germany).

DNA-FISH imaging was performed using a Dragonfly spinning disk confocal system (Andor Technology PLC, Belfast, United Kingdom) connected to an inverted Leica Dmi8 microscope (Leica Microsystems CMS GmbH). The signals were detected simultaneously by two sCMOS Zyla (Andor) 2048 × 2048 cameras, 2 × 2 binning, CF40 pinhole, and 12-bit depth. Images were acquired with a 63×/1.3 glycerol objective and excited with 4 different laser lines (two channels per camera): 405, 488, 561 and 637 nm laser lines.

### 2.6. Image Analysis

Arivis Vision4D 3.1.2-3.4 was used for image visualization and analysis. Quantitative immunofluorescence analysis was performed with a dedicated pipeline that automatically segmented multiple nuclei per stack in 3D, using denoising and Otsu threshold operation on the independent Hoechst channel. The automated analysis restricts signal quantification only to the nucleus by segmenting the nuclear borders from the full z-stack, therefore eliminating the effects of cytoplasmic background on the mean nuclear fluorescence intensity. Nuclear volumes and total fluorescent intensities of the epigenetic marks and the DNA were calculated and exported to MATLAB R2019b (MathWorks) for further analysis.

Quantification of live H3K27me3 puncta was performed by first assessing the signal intensity profile of each nucleus and subtracting the nuclear background intensity for each nucleus individually, preserving only the bright repressive puncta. Individual puncta were then automatically segmented in 3D using Otsu operation, and puncta below volume of 0.01 µm^3^ were filtered out. Radial 3D chromatin distribution analysis was performed as previously described [30] by generating consecutive radial shells within the segmented nucleus and quantifying the mean H2B-GFP intensity within the shell divided by the mean cytoplasmic intensity.

### 2.7. Computational Model

The coarse-grained model simulates the 176.2 Mbps-long *Drosophila* genome with N = 35,240 beads connected by springs to model each chromosome, where a bead of diameter σ=30 nm includes 5 kbps of DNA. We confined four such flexible polymers, which represent the chromosomes of *Drosophila*, to a sphere which represents the nucleus. To model the relative lengths of the euchromatin, PCH and H3K27me3 blocks, we analyzed the experimental chip sequence data at 5 Kbps resolution and converted it into beads units [31,32]. According to these data, 70% of *Drosophila* genome regions are euchromatic and 30% are identified as PCH [31]. The chip-seq data of H3K27me3 markers are currently only available for euchromatin. The data indicate that 22% of the euchromatin regions [32] have H3K27me3 modifications. In addition, the data show that the average patch length for H3K27me3 is 17 beads (or 85 Kbps) and that the variation in patch length has an exponential distribution. The H3K27me3 modifications in PCH regions are very abundant [33,34], and we assume 75–80% of the PCH is covered with H3K27me3. We therefore estimate that 40% of the *Drosophila* genome contains H3K27me3 markers based on its percentage in euchromatin and PCH. Using these values, we generated beads identified as H3K27me3 patches from an exponential distribution with a mean length of 17 beads, which covers 40% of the genome. These H3K27me3 patches were generated randomly along the genome using a Monte Carlo method [35], which ensures that two patches cannot overlap. Within the spherical confinement (modeled as an impenetrable wall), the chromatin volume fraction is taken to be 15% [36]. The nuclear lamina is modeled as a thin layer of another type of unconnected beads (lamina beads) which are localized at the surface of the spherical confinement volume. For convenience, we take the lamina beads to be fixed in position and to have the same size as the chromosomal beads. A truncated Lennard-Jones (LJ) potential [35] accounts for bead attractions if the distance between two beads is within 2.5σ, along with a short-distance, strong repulsion which prevents them from overlapping. The attraction between beads is motivated by our previous experimental observations and simulations of chromosome phase separation in live *Drosophila* nuclei [35]. In our model, euchromatin and heterochromatin (PCH and H3K27me3) are distinguished by the strength of their Lennard-Jones attraction (ε_i j_ where i j can be E (euchromatin) or H (heterochromatin). Heterochromatin beads ($\varepsilon_{HH}=0.5\;k_{B}T)$, which are more condensed in the nucleus, attract each other more strongly than euchromatin beads $\left(\varepsilon_{EE}=0.35\;k_{B}T\right)$. However, each bead type is self-attractive, since both types of chromatin are observed to be phase separated from the nucleoplasm; in our model, the chains from the nucleoplasm are no longer in good solvent and collapse for $\varepsilon =0.3\;k_{B}T$ [37]. A bead of H3K27me3 forms a bond with a nearby lamina bead when their separation is within 1.5σ. Each H3K27me3 bead is restricted to form a maximum of one bond with the lamina beads. After such a bond is formed, the distance between the H3K27me3 and lamina beads can fluctuate, and we use a harmonic (spring-like) potential with a spring constant K to account for the change in the bond energy as a function of the H3K27me3-lamin bead separation relative to their optimal spacing of σ. These bonds are broken when the fluctuating distance between beads exceeds 2.5σ. A detailed description of the dynamic harmonic potential (bonding/unbonding) can be found in Refs. [35,38]. We used the LAMMPS molecular dynamics package to simulate our model system [39].

### 2.8. Statistics

Quantitative immunofluorescence parameters from control and *SUN/koi* mutant nuclei were compared using a mixed linear model, with genotype as a fixed effect and with larva and muscles as random effects [40]. The number of larvae and nuclei for each group and experiment are listed in the legends. Statistics were performed in R v.4.0.2 using the package lmerTest v.3.1-2. Linear fits to the epigenetic mark intensities, and total chromatin as function of the nuclear volume were generated using linear mixed-effects model fit. BoxPlotR was used to generate box plots [41], in which the center lines represent the medians, box limits indicate 25th and 75th percentiles, and whiskers, determined using Tukey’s method, extended to data points <1.5 interquartile ranges from the first and third quartiles as determined by the BoxPlotR (http://shiny.chemgrid.org/boxplotr/, accessed on 17 March 2022). 

## 3. Results

### 3.1. Genome-Wide Polycomb, HP1a and RNA-Pol II Binding Profiles Reveal Increased Repression and Reduced Activation in SUN/koi Mutant Muscle Fibers

Although the LINC complex has been implicated in the regulation of the epigenetic state, chromatin organization and overall gene expression [4], it is not clear whether and how LINC mediated mechanotransduction is involved in the downstream genome-wide chromatin repression and activation. Here, we investigated the contribution of the LINC complex to the epigenetic chromatin state by analyzing the phenotype of *SUN/koi* mutant muscles in *Drosophila* third instar larvae. Whereas the two Nesprin-like genes of *Drosophila* have additional functions in cells, due to non-KASH containing isoforms, *SUN/koi*, the only SUN representative in *Drosophila* does not appear to exhibit LINC-unrelated functions [20,42]. We therefore investigated changes to genome-wide chromatin repression and activation states in the fully differentiated muscles of *SUN/koi* mutants in vivo using the Targeted DamID (TaDa) technique that allows tissue-specific profiling of DNA-binding proteins in the intact living organism [43]. Here, we compared the DNA-binding profiles in *SUN/koi* mutated and control muscle fibers of two chromatin factors: (1) Polycomb, the H3K27me3 reader representative of the polycomb-group associated heterochromatin, (2) HP1a, the H3K9me3 reader representative of the HP1a-associated heterochromatin, and in addition, the profile of RNA-Pol II subunit RpII215, representative of actively transcribed euchromatin. Polycomb, the subunit of the Prc1 complex Cbx8, HP1a, and RpII215, the main catalytic subunit of RNA-Po II, fused to the DNA adenine methyltransferase (Dam) were driven to *Drosophila* larval muscles using *Mef2-GAL4* driver in a temporally controlled manner (by combined expression of the temperature sensitive inhibitor *Gal80^ts^*), inducing expression of the Dam-fusion proteins only at the third instar larval stage, when muscles are mature and fully differentiated. The experiment included *n* = 3 independent replicates of Dam–Polycomb, Dam-HP1a, Dam-RNA Pol II, and Dam only (as reference), for control and for *SUN*/*koi* nutant genotypes, with *n* = 25 larvae in each group. Following 10 h of Dam expression, larvae were dissected, DNA was extracted and digested with DpnI at the Dam methylated GATC sites. Further amplification, processing, and bioinformatics analysis were performed as described [27]. 

To compare gene binding profiles between *SUN*/*koi* and control muscle fibers, we performed a PCA regression analysis on Dam-Polycomb, Dam-HP1a, and Dam-Pol II, normalized to Dam binding alone (Figure 1A). The cut-off criteria for significant change in gene binding between *SUN/koi* and control was set to false discovery rate (FDR) < 0.05, z-score > 1.96, and GATC sites > 1. The gray dots in Figure 1A represent genes with significantly altered binding, such that genes below the regression line represent increased binding in the *SUN/koi* mutant, and genes above the regression line represent decreased binding in *SUN/koi* compared to control. Overall, we detected significantly altered binding to 148 genes in the Polycomb group, 173 genes in the HP1a group and 206 genes in the RNA-Pol II group. The full list of identified genes for each group is provided in the Appendix A. The full DamID data, including the non-significantly altered genes, are openly available in Mendeley Data at DOI: 10.17632/czjj5m7btr.1 https://data.mendeley.com/datasets/czjj5m7btr/1 (accessed on 17 March 2022). Note the robust increase in Polycomb binding observed in the *SUN/koi* mutant, where 143 out of 148 significant genes showed enhanced Polycomb occupancy. Notably, there was only 0.5 % and 2.4% overlap between the Polycomb-RNA-Pol II and HP1a-RNA-Pol II groups, respectively, and 6.3% overlap between the Polycomb-HP1a groups. Figure 1B illustrates representative examples of non-overlapping occupancy of Dam–Polycomb, Dam-HP1a, and Dam-Pol II in 3 genes expressed in the larval muscle fibers, namely alphaTub84B (left), Nup62 (middle) and Act57B (right) genes. Interestingly, each of these genes showed a statistically significant altered occupancy in *SUN/koi* mutant muscles for only one chromatin factor (red boxes) with increased Polycomb binding to alphaTub84B, increased HP1a binding to Nup62 and decreased RNA-Pol II binding to the Act57B gene. The corresponding Zpca scores are listed for each gene on Figure 1B.

We further paralleled genome-wide changes in Polycomb, HP1a and RNA-Pol II occupancy in the *SUN/koi* mutant muscles. Figure 2A depicts a heatmap of (*SUN/koi* versus control) fold change occupancy for each chromatin factor, such that red indicates increased occupancy and blue indicates decreased occupancy. Non-significant genes are labeled in gray. Gene clustering was defined by k-means, and the number of clusters was estimated with Gap statistics. Overall, we observe minimal overlap between the group of genes that showed altered binding to the three factors, namely, Polycomb, HP1a, and RNA-Pol II, as the majority of genes colored in each group are non-significant (gray) in the others. Cluster 1 in Figure 2A includes genes predicted to undergo decreased transcription. It contains the genes that showed increased Polycomb occupancy, increased HP1a binding and decreased RNA-Pol II binding. Cluster 2 was predicted to undergo reduced transcription due to decreased RNA-Pol II binding. These genes partially overlapped with reduced HP1a occupancy genes, possibly representing genes that are normally positively regulated by HP1a [44]. Cluster 3 includes three non-overlapping groups of genes: those exhibited increased RNA-Pol II binding, and therefore were expected to be upregulated, and two groups of genes that showed increased or decreased occupancy by HP1a. The latter two groups might represent genes that are normally either repressed, or activated by HP1a, and in the mutant are affected in the opposing direction. Further, the three clusters showed non-overlapping functionalities according to GO analysis (Figure 2B). Significantly altered hit genes in the *SUN/koi* mutant corresponded to distinct functions. Cluster 2, predicted to include genes that were transcriptionally repressed, included genes associated with muscle function, e.g., genes coding for contractile proteins and for myosin complexes. It also included genes associated with pupal adhesion, predicted to be expressed by salivary gland cells, possibly due to leaky expression of Mef2-GAL4 in the salivary gland. Notably, not all of the genes that increased binding to RNA Pol II are predicted to be transcribed, as RNA Pol II transcription might be paused. Cluster 1, predicted to include genes that were also transcriptionally repressed, contained genes coding for components of mTOR signaling involved in muscle cell growth [45]. Figure 2C includes a list of specific genes of interest, and their fold change values. Among them are 26 muscle-specific genes mostly predicted to be transcriptionally downregulated in the *SUN/koi* mutant muscles. Such transcriptional repression explains the significantly thinner muscle fibers and impaired contraction of the *SUN/koi* mutant muscles. Interestingly, we also observed decreased binding of RNA-Pol II and increased binding of HP1a to a group of genes involved in nuclear transport, predicting impaired functionality of the muscle fibers as well. In addition, a group of transcriptional regulators showed increased Polycomb binding and decreased RNA-Pol II binding, again predicting impaired function of the muscle fibers, and explaining the weakness of the larval muscles and their thinner phenotype, resulting in slower larval movement.

Taken together, our genome-wide analysis indicates that defective LINC complex function associates with increased chromatin repression, mostly through increased Polycomb binding, and decreased transcriptional activation through reduced RNA-Pol II binding.

To test if HP1a, RNA-Pol II, and Polycomb upregulated hits tend to cluster more than expected by chance, we ran a simulation in which we randomly picked genes from the *Drosophila* dm6 genome (chr2R, chr2L, chr3R and chr3L) and counted the number of genes that are in a distance <10,000 bp (expected number of close genes). To match the HP1a-, RNA-Pol II- and Polycomb-upregulated hits distributions over the genome, we picked the same number of genes, corresponding to the upregulated positive hits for each group. The process was repeated Nruns = 1000 times and the *p*-values were then calculated for the fraction NGE + 1/Nruns + 1, where NGE is the number of simulations with a value greater than or equal to the number of hits for each factor (Figure 2D). This analysis indicates that the Polycomb-upregulated genes tend to cluster along the chromosomes significantly more compared to randomly distributed genes; namely, 14 Polycomb-upregulated genes were located within 10kbp compared to a median of 7 from Monte Carlo simulation (*p* = 0.012, number of gene hits *n* = 136). The number of upregulated gene hits for RNA-Pol II and HP1a was significantly lower; however, a similar tendency for clustering was observed, where nine RNA Pol II upregulated genes were located within 10 kbp compared to a median of five from Monte Carlo simulation (*p* = 0.002, and *n* = 75) and nine HP1a upregulated genes clustered within 10bp compared to a median of five from Monte Carlo simulation (*p* = 0.001, *n* = 65). Similar analysis of the distribution of RNA Pol II downregulated genes did not show such a clustering tendency (Appendix A). This analysis indicated a change in chromatin distribution in the *SUN/koi* mutants where Polycomb hits tend to cluster along the chromosome’s length.

### 3.2. Perturbed LINC Complex Associates with Increased Epigenetic Chromatin Repression and Reduced Epigenetic Chromatin Activation in Muscle Nuclei

The changes in Polycomb and RNA-Pol II genome binding led us to address possible changes in the meso-scale epigenetic landscape in *SUN/koi* mutated muscle fibers of third instar larvae, using antibody staining for three epigenetic modifications, namely H3K9ac, H3K27me3 and H3K9me3. Figure 3A shows representative muscle nuclei labeled with the repressive chromatin modifications H3K27me3 (purple) and H3K9me3 (green), with overall increased signal of both marks, in the *SUN/koi* mutated muscle nuclei. Figure 3B shows quantification for the mean nuclear fluorescence intensity with a 43% increase in H3K27me3 (*p* < 0.05) and an 82% increase in H3K9me3 (*p* < 0.01). The overall increased intensity of repressive chromatin marks is in agreement with the increased Polycomb and HP1a binding observed with the targeted DamID. As previously reported, the *SUN/koi* mutant displays smaller and variable myonuclear size and increased DNA ploidy [46]. We therefore asked whether the observed epigenetic changes correlate with the nuclear size. Figure 3C shows the mean H3K27me3, and H3K9me3 fluorescence intensity, for each nucleus, as a function of the nuclear volume (log10 scale). Interestingly, both of the repressive marks displayed increased repression with smaller nuclear volumes. We performed linear mixed model analysis (with genotype as a fixed effect and with larva and muscles as random effects) on the dependence of mean H3K27me3 intensity (left), and mean H3K9me3 intensity (right) on nuclear volume in the *SUN/koi* group compared to the control (red and black dots, respectively). We first compared fitted mixed models for all pulled nuclear volumes and found significant differences between the *SUN/koi* and control groups for H3K27me3 and H3K9me3 repressive marks (*p* < 0.01). To assess the contribution of the smaller nuclei, which are present only in the *SUN/koi* groups, we then fitted mixed models only for the larger nuclei, with overlapping volumes between the *SUN/koi* and control groups. The larger, overlapping volume nuclei showed no significant difference between the *SUN/koi* and control models for both repressive marks (*p* > 0.2), suggesting that the smaller nuclei in the *SUN/koi* groups are the major contributors to the increased H3K27me3 and H3K9me3 repression. To address whether the increased repressive epigenetic modifications represent muscles lacking LINC function, we characterized larval mutant muscles deficient in both *Msp300* and *klar* using a double mutant combination of *klar;Msp300* lacking the KASH domain (ΔKASH) [47]. We found a similar tendency in the ΔKASH mutants (Appendix A), with a 48% increase in mean fluorescent H3K27me3 intensity (*p* < 0.001) and a 25% increase in mean fluorescent H3K9me3 intensity (*p* < 0.001) signals, and the repressive mark intensity inversely correlated with nuclear volume (Appendix A).

*Drosophila* muscle nuclei are polyploid, and we showed previously that chromatin volume scales linearly with nuclear volume, maintaining a constant chromatin volume fraction constant [30]. The inverse correlation of chromatin repression with nuclear volume, observed in the *SUN/koi* mutant muscles, suggested increased chromatin condensation, which might contribute to the excessive chromatin repression [48]. To address this, we compared the global (nuclear scale) DNA condensation in *SUN/koi* and control myonuclei. Figure 3D shows a plot of the total DNA intensity as a function of nuclear volume (log10 scale) for *SUN/koi* and control groups (red and black dots, respectively). A linear mixed model analysis confirms a significant difference between the *SUN/koi* and control fits (*p* < 0.01) with a leftward-shift in the *SUN/koi* total DNA–volume relationship, indicating that for each *SUN/koi* nucleus, the DNA is condensed in a smaller nuclear volume relative to the control. Interestingly, we did not observe a similar trend of DNA condensation in the ΔKASH mutant muscle fibers (Appendix A).

### 3.3. Increased H3K27me3 Repression Is Specific for Mature LINC Mutant Muscles

We further investigated whether the increased H3K27me3 repression in the muscles is caused by the requirement for *SUN/koi* in the course of muscle differentiation during embryonic stages, or if it is also observed in fully differentiated muscle fibers. To achieve this, we conditionally knocked down *SUN/koi* by RNAi in larval muscle fibers, after muscle differentiation had been fully completed, utilizing the temperature-sensitive *GAL80/Mef2-GAL4* driver. Whereas muscle-specific *SUN/koi* knockdown induced after muscle differentiation (in hatched larvae) did not cause significant defects in nuclear positioning or sarcomere arrangement, nor did it affect nuclear volume, it did recapitulate the increased levels of H3K27me3 modification with a 45% increase in the mean nuclear fluorescent intensity (Figure 4A,B, *p* = 0.011). Consistently, the temporal *SUN/koi* knockdown in differentiated muscles showed no dependance of H3K27me3 levels on nuclear volume (Figure 4C) and no change in the total DNA/nuclear volume relationship (Figure 4D). These results suggest that while the *SUN/koi* is required to maintain nuclear volume, optimal DNA density, and nuclear positioning during embryonic development, proper LINC function in mature muscles is required to inhibit epigenetic repression, independently of changes in DNA density.

Since the LINC complex is required for the cytoplasmic–nuclear force transmission, which is particularly high in mature muscle fibers, we tested whether alternative perturbation of such forces will show a similar increase in repressive chromatin. To this end, muscle-specific, temporal disruption of the microtubule (MT) network, normally exerting high compressive forces on the nucleus, was induced by conditional overexpression of spastin, a MT severing protein. This treatment disrupted the MT in muscles for a short time period without affecting sarcomere organization or muscle size [46]. Figure 4E depicts representative control and spastin overexpression muscle nuclei, with preserved H3K27me3 and DNA density upon disruption of the MT network, despite increased nuclear volume. We compared the H3K27 tri-methylated fluorescent signal and demonstrate that the mean fluorescent intensity of H3K27 tri-methylation (Figure 4F), as well as the ratio between mean H3K27 tri-methylation fluorescent signal and nuclear volume (Figure 4G) in spastin-treated muscle nuclei, did not differ from control. Figure 4H quantifies the proportional increase in nuclear volume and total DNA content with spastin overexpression (presumably as a result of DNA endoreplication), implying that DNA/nuclear volume relationships were similar to control. This result reinforces the specific and unique role of the LINC complex in inhibition of epigenetic repressive modification in the mature muscle fibers.

### 3.4. Reduced Epigenetic Gene Acetylation Is Observed in the SUN/koi Mutant Muscles

In agreement with the reduced RNA-Pol II binding to muscle-specific genes, we identify a decrease in H3K9ac chromatin modification, which marks the active promotors in the *SUN/koi* mutant muscle fibers (Figure 5A). Figure 5B quantifies the mean nuclear H3K9ac intensity with 37% reduction in the *SUN/koi* mutant (*p* < 0.01). Notably, the active H3K9ac mark showed no correlation with nuclear volume (Figure 5C). Interestingly, the ΔKASH mutants did not show a similar tendency to the *SUN/koi* mutants, as we observed no significant change in the mean nuclear H3K9ac intensity (*p* = 0.19) (Appendix A). Overall, these data indicate a reduced active landscape in the *SUN/koi* that is consistent with the reduced binding of RNA Pol II to an array of genes, including muscle genes.

### 3.5. Live Imaging of the 3D Distribution of H3K27me3 Sites Reveals Increased Cluster Size in *SUN/koi* Mutated Muscle Fibers

To investigate the spatial distribution of H3K27me3 sites in the *SUN/koi* mutant muscle nuclei, we performed live 3D imaging of larval muscles utilizing the H3K27me3-GFP mintbody [49]. Representative 3D spatial distribution of H3K27me3-GFP in control and *SUN/koi* mutated nuclei is shown in Appendix A, respectively. Figure 6 shows middle confocal sections of representative control (left) and *SUN/koi* mutant (right) muscle nuclei labeled with H3K27me-GFP mintbody. The strong H3K27me3 signal appears punctuated within the nucleus, with weaker nuclear and cytoplasmic background. Compared to control distribution, the H3K27me3 puncta in the *SUN/koi* mutant nuclei appeared more spread out throughout the chromatin, resulting in larger repressive H3K27me3 hubs. We quantified the spread of the puncta by measuring the volume (Figure 6B) and the number (Figure 6C) of strong H3K27me3 puncta in each group. The *SUN/koi* puncta volume was, on average, 149% higher than the control (*p* < 0.01) with no significant change in the number of the puncta between the groups. Thus, the live 3D distribution of H3K27me3 puncta demonstrates increased spreading of this repressive mark to neighboring chromatin regions in the *SUN/koi* mutated muscle nuclei, further supporting the clustering of positive Polycomb binding location across the genome (Figure 2D). Considering the larger H3K27me3 puncta in live *SUN/koi* mutant muscle nuclei, together with the increased Polycomb binding (Figure 1A) and the clustering of positive Polycomb binding sites (Figure 2D), our findings suggest that *SUN/koi* disruption induces increased repression via the spread of repressive H3K27me3 chromatin modifications to nearby chromatin regions.

### 3.6. Dissociation of H3K27me3 Chromatin from the Lamina Is Linked to Increased H3K27me3 Clustering in LINC Mutant Muscle

Our previous work indicated peripheral chromatin organization in the nuclei of mature muscle fibers, which is sensitive to lamin A/C levels, when imaged live, in vivo [30]. To assess the meso-scale chromatin organization in *SUN/koi* mutated muscles, we imaged live larval myonuclei co-labeled with H2B-mRFP and Klar-GFP. However, in the *SUN/koi* mutant, the Klar-GFP did not localize to the nuclear envelope due to the lack of *SUN/koi* (Appendix A). Nevertheless, we analyzed the live, 3D distribution of chromatin by quantifying H2A-GFP density in radial shells from the nuclear periphery to the nuclear center (Figure 7A). *SUN/koi* mutated muscle nuclei exhibit altered chromatin distribution with a reduction in peripheral chromatin density and a shift towards the center.

Experimentally, the live 3D analysis in this study suggests that disruption of the LINC complex results in excessive H3K27 tri-methylation, chromatin redistribution from the periphery towards the center of the nucleus, and extra Polycomb binding to genes hubs, presumably leading to increased size of repressive H3K27me3 clusters. We therefore studied a computational model to explore a possible link between dissociation of the chromatin, including tri-methylated H3K27 hubs, from the nuclear envelope and the formation of larger repressive clusters in the nucleoplasm of LINC mutants. To this end, we introduced a coarse-grained model for the *Drosophila* genome that incorporates binding/unbinding of H3K27me3-modified chromatin (facultative heterochromatin) to the lamina. The *Drosophila* genome was simulated by beads connected by springs to model each chromosome, where a bead of diameter σ=30 nm includes 5 kbps of DNA. We confined four such flexible polymers, which represented the chromosomes of *Drosophila*, to a sphere which represented the nucleus. The chromosomal chains consisted of three types of beads: euchromatin beads, pericentromeric heterochromatin (PCH) beads and H3K27me3-modified beads (Figure 7B, red, blue and yellow beads, respectively). In our model, euchromatin and heterochromatin (PCH and H3K27me3) are distinguished by ε-the strength of their Lennard-Jones self-attraction (LJ, see further details in Section 2). H3K27me3 beads form bonds with nearby lamina beads with a fluctuating H3K27me3-lamin bead distance after bond formation. The change in the bond energy as a function of the H3K27me3-lamin bead separation is described by a harmonic (spring-like) potential with a spring constant K. Our simulation results show that decreasing the spring constant K, namely the H3K27me3-lamina bond strength, increases the average H3K27me3 cluster size in the nucleoplasm, resulting in a chromatin density shift from peripheral to more central (Figure 7B). This occurs because smaller values of K lead to increased probability of H3K27me3-lamin unbinding. The H3K27me3 that has unbonded from the lamin phase separates in the nucleoplasm due to the attractive interactions of the H3K27me3 beads; this increases the chromatin volume in the nucleoplasm relative to periphery. Overall, our experimental and computational results suggest a model whereby disruption of the LINC complex in the mature muscle weakens the interactions of repressive chromatin regions with the lamina, resulting in increased repressive H2K27me3 clustering in the nucleoplasm and an overall chromatin shift from the periphery towards the center (Figure 7C). Functionally, the increased epigenetic repression points to the downregulation of muscle contractile genes in agreement with thinner muscle and perturbed mobility of the mutant larvae.

## 4. Discussion

The LINC complex has been implicated in the regulation of chromatin 3D organization; however, its specific contribution and the underlying mechanism are yet to be elucidated. Here, we demonstrate that the LINC complex is required to reduce chromatin repression in the mature skeletal muscle. The genome-wide binding analysis of a set of transcription regulators, including Polycomb, HP1a and RNA-Pol II, performed in *SUN/koi* mutant muscles, predicted an overall enhanced transcription repression. Consistently, increased levels of the repressive tri-methylated H3K27 and H3K9 and decreased levels of the active H3K9ac chromatin modifications were observed. Whereas DNA condensation is elevated in *SUN/koi* mutant muscles due to reduced nuclear volume, we show that *SUN/koi* conditional knockdown nuclei with nuclear size comparable to control still exhibit enhanced epigenetic repression, indicating a mechanism that is not based on DNA condensation. Remarkably, live imaging of tri-methylated H3K27 marked chromatin in *SUN/koi* mutant muscles reveals larger repressive chromatin clusters that correlate with a transition of the chromatin from the nuclear periphery towards the nuclear center. Computational modeling of the distribution of H3K27me3 sites indicates that dissociation of chromatin–nuclear lamina binding leads to the formation of larger H3K27me3 repressive clusters, consistent with our observations. Mechanistically we suggest that weakening of chromatin binding to the nuclear lamina in the *SUN/koi* mutants leads to excessive clustering of H3K27me3 sites and to spreading of Polycomb binding to neighboring genes. Taken together, these results suggest that the LINC complex is required for inhibition of excessive chromatin epigenetic repression necessary for robust transcriptional regulation in fully differentiated muscle fibers.

Among the polycomb hits (Cluster 1), we identified enrichment for two GO terms functionally associated with TOR signaling, a major signaling pathway regulating muscle growth. Enhanced transcriptional repression of TOR signaling in the *SUN/koi* mutants is consistent with the thinner muscle phenotype observed in these muscles [20,46]. Since Polycomb binds specifically to tri-methylated H3K27-marked chromatin, the enhanced Polycomb binding is consistent with the increased H3K27me3 fluorescent labeling observed in the *SUN/koi* mutant muscles. At this stage, we cannot determine whether increased polycomb binding leads to increased H3K27me3 or vice versa, since Polycomb was demonstrated to self-aggregate with chromatin due to phase separation [50], but also to further promote tri-methylation of H3K27 [48]. Our observation that in muscles of live *SUN/koi* mutant larvae the H3K27me3-positive puncta grew in size is consistent with increased self-aggregation of pre-existing H3K27me3-associated chromatin clusters.

Polycomb is a member of polycomb repressive complexes (PRCs) and has been extensively studied in development and differentiating cells. PRC binding to epigenetically modified chromatin leads to a transcriptionally poised state, which then prevents unscheduled differentiation [48]. Therefore, enhanced binding of Polycomb to chromatin might inhibit transcriptionally regulated developmental transitions in the larval muscle fibers. The idea that such transitions might be regulated by mechanical signals, mediated by the LINC complex, is intriguing and novel. Consistent with the contribution of the LINC complex to the control of developmental transitions is a precocious differentiation of differentiating keratinocytes deficient in both SUN1 and SUN2 [17], where lack of SUN proteins resulted in reduced repression, higher accessibility of the chromatin and activation of terminal keratinocyte differentiation. The different outcome might stem from a distinct state of cellular differentiation and the chromatin state in each of the experimental systems [51]. Importantly, however, the LINC complex appears to be essential for preservation of the epigenetic state of the chromatin in both cases.

A number of studies suggested a link between nuclear mechanical stimulation and chromatin epigenetic repression. For example, long-term stretch of epidermal progenitor cells led to increased H3K27me3 occupancy, resulting in silencing of epidermal differentiation [52,53]. Furthermore, mechanical inputs inducing nuclear deformation led to changes in the distribution of heterochromatin and euchromatin in various cell types, including cardiomyocytes and keratinocytes [4,15]. Mechanistically, it has been proposed that nuclear deformation changes the extent of chromatin binding to the nuclear envelope, affecting primarily repressed chromatin [4]. Interestingly, epidermal progenitor cells subjected to mechanical stretch exhibited reduced H3K9 tri-methylation associated with HP1a binding, which was accompanied by a concomitant increase in PRC2 binding and enhanced global repression [53]. Our analysis of LINC mutant muscles did not reveal similar replacement between the binding of Polycomb and that of HP1a and suggests that each of these transcription repressors changed its binding independently of the other on mostly distinct set of genes.

The RNA-Pol II binding profile in *SUN/koi* mutant muscles reveals genes in Cluster 2 with decreased binding to the transcriptional machinery complex. According to GO analysis, this cluster contains a group of genes coding for muscle contractile proteins. Reduced transcription of such genes is expected to result in muscle weakening and failure in larval locomotion, and indeed, both phenotypes were observed in the *SUN/koi* mutants. Notably, because the DamId analysis was performed at a late stage of third instar larvae, some of the muscle-specific genes undergo transcription reduction due to the transition from larvae to pupal stage (flybase.org). For example, muscle-specific genes, including myosin heavy chain (MHC), *sallimus* (*Drosophila* titin-like gene), *MSP300* (Nesprin-like gene), *alpha actinin (Actn)* and others, show a significant reduction in mRNA levels in the late third instar larval stage. In such genes, alterations of RNA-Pol II binding in the mutant muscles might not be detectable. Nevertheless, a group of other muscle genes did show significantly reduced binding to RNA-Pol II, e.g., *myofilin* (*Mf*), *tropomyosin1* (*Tm1)*, *myosin light chain* (Mlc), *actin 57B* (Act57B), *troponinC73F* (TpnC73F) and others. Consistently, antibody staining indicated reduced TpnC protein levels (Appendix A). In addition, a distinct group of muscle genes showed increased Polycomb binding (e.g., *Zasp52, Amph, betaTub56D* and others), predicting their increased repression in the *SUN/koi* mutant muscles. Lastly, transcription itself might be attenuated due to the increased binding of Polycomb and HP1a to components of the transcription machinery, such as mediator complex proteins *med10* and *med28*. The decreased levels of H3K9 acetylation in the *SUN/koi* mutant muscles are consistent with the paralleled reduction in RNA-Pol II binding to genes in Cluster 2. Importantly, the lack of overlap between the group of genes that showed increased Polycomb binding (Cluster 1) and the group of genes that exhibited decreased RNA Pol II binding (Cluster 2) suggests that although both processes are regulated by the LINC complex, they might result from distinct effects of this complex on repressed and active chromatin regions.

Although enhanced DNA condensation and compaction could explain the upregulation of epigenetic repression in the LINC mutant nuclei, we do not favor this explanation, since we found that elevated epigenetic repression is also observed in nuclei of normal size with conditional *SUN/koi* knockdown. The simulation results described in Figure 7B provide an alternative mechanism whereby repressive H3K27me3 clusters grow in size in the nucleoplasm if chromatin affinity to the nuclear lamina is reduced compared to the wildtype situation. Experimentally, we show that chromatin distribution indeed shifts from the nuclear periphery towards the nuclear center in the *SUN/koi* mutant nuclei, implicating weakening of chromatin–nuclear lamina binding. Consistently, recent studies indicated that in LINC mutants, the LEM domain protein Emerin which, together with BAF, promotes chromatin tethering to the lamina, fails to oligomerize at the inner nuclear membrane [54], and in line with this result, BAF is eliminated from the nuclear envelope in LINC mutant muscles [55]; both factors predict chromatin dissociation from the nuclear envelope in the LINC mutants, as we observed. Therefore, we favor a mechanism by which the LINC complex stabilizes chromatin association with the nuclear envelope, preventing excessive epigenetic repression.

In summary, this study analyzed the contribution of the LINC complex to chromatin epigenetics and the binding of transcriptional regulators in fully differentiated muscle fibers. We demonstrate that in the absence of functional LINC complex, chromatin epigenetic repression is elevated, mainly due to enhanced binding of Polycomb and excessive clustering of repressed chromatin in the nucleoplasm, while Pol II binding to muscle-specific genes decreases. This forms the basis for abrogated muscle structure and function in LINC mutant observed in striated muscles, as well as in LINC-associated cardiomyopathies.

## Figures and Tables

**Figure 1 cells-12-00932-f001:**
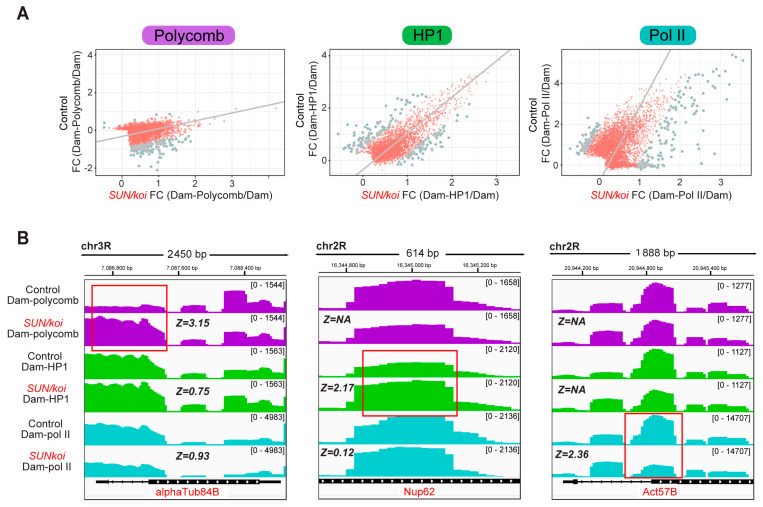
Increased Polycomb and altered HP1a and RNA-Pol II occupancies in LINC mutated muscle fibers. (**A**) PCA regression on Dam-Polycomb (Cbx8, a component of the Prc1 subcomplex, (**left**)), Dam-HP1a (**middle**), Dam-Pol II (RpII215, the catalytic subunit of RNA-Pol II, (**right**)) gene occupancy, normalized to Dam alone, in control versus *SUN*/*koi* mutated muscle fibers. Gray dots represent genes with significantly altered binding profile when cut-off criteria are set to FDR < 0.05, Z-score > 1.96, GATC sites > 1. (**B**) Dam-Polycomb, Dam-HP1a, and Dam-Pol II, occupancy, normalized to Dam alone, for the alphaTub84B (**left**), Nup62 (**middle**), and Act57B (**right**) genes. Each gene presents a statistically significant altered occupancy in only one chromatin factor upon *SUN*/*koi* mutation (highlighted with red box) and the corresponding Zpca scores are listed for each gene.

**Figure 2 cells-12-00932-f002:**
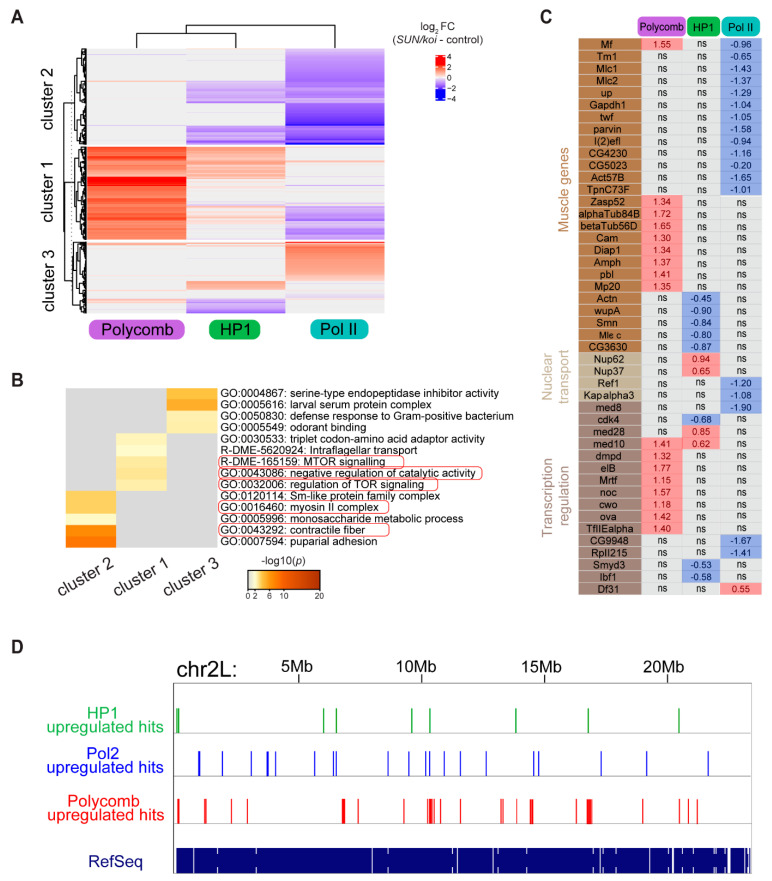
Polycomb, HP1a and RNA-Pol II binding regulate distinct group of genes in the LINC mutated muscle fibers, with overall increased Polycomb repression and decreased RNA-Pol II activation. (**A**) The difference in *SUN/koi* and control fold change occupancy represented as heatmap for significantly altered genes in the Polycomb, HP1a, and RNA-Pol II groups. Non-significant genes are labeled in gray. K-means gene clustering predicts downregulated genes in Cluster 1 (increased Polycomb binding), and Cluster 2 (decreased RNA-Pol II binding), and heterogenous genes in cluster 3. (**B**) GO enrichment analysis (heatmap of *p*-values) on the three clusters of genes identified in (**A**). (**C**) The difference in *SUN/koi* and control fold change occupancy for specific genes of interest; ns is not significant. (**D**) Representative image of the distribution of the upregulated HP1a, RNA-Pol II and Polycomb hits on chromosome 2 L from DamID analysis, indicating increased clustering of Polycomb-upregulated genes along the genome.

**Figure 3 cells-12-00932-f003:**
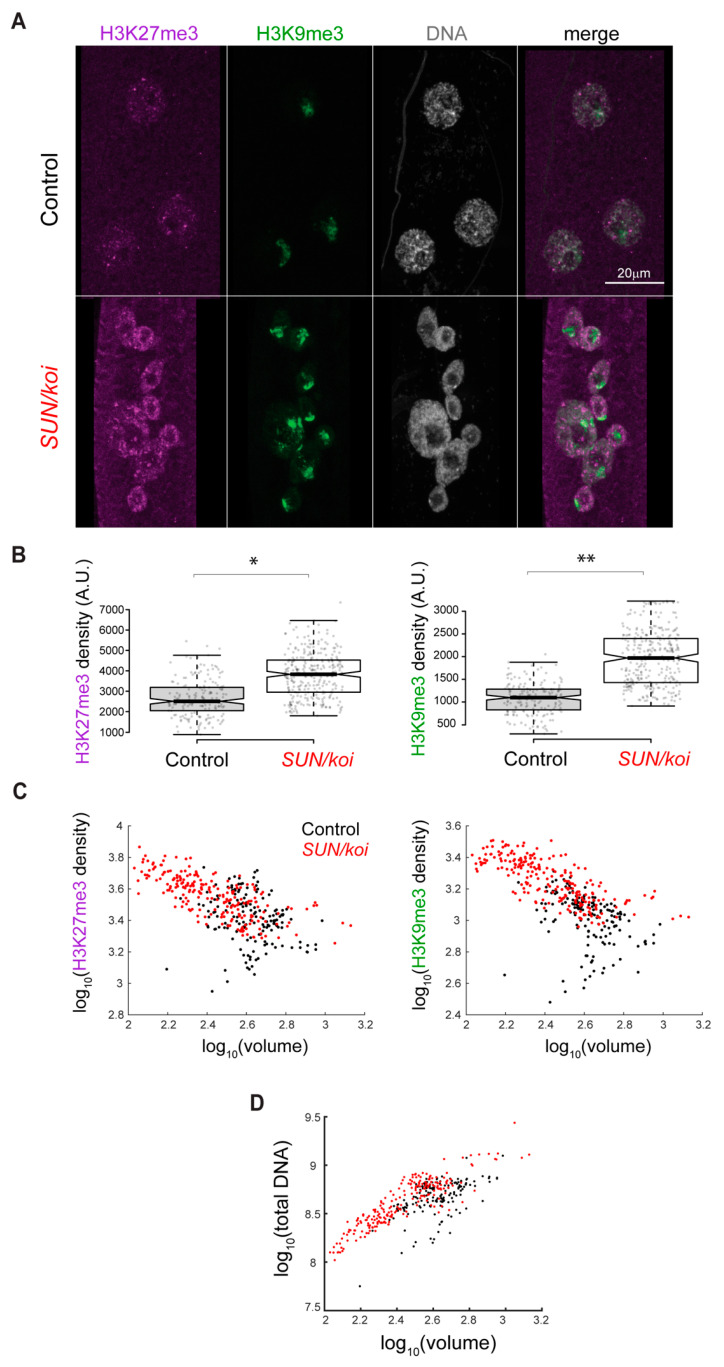
Increased repressive H3K27me3 and H3K9me3 chromatin density, inversely correlated with nuclear volume, in *SUN/koi* mutated muscle fibers. (**A**) Muscle nuclei labeled with H3K27me3 (purple), H3K9me3 (green) and Hoechst for total DNA (gray). (**B**) Quantification of mean nuclear fluorescence intensity shows increased H3K27me3 and H3K9me3 in *SUN/koi* mutated muscle fibers. (**C**) Mean nuclear fluorescence intensity is plotted against the corresponding nuclear volume (log10 scale) for each epigenetic mark. Significant difference in the linear mixed model fit between the *SUN/koi* and control groups, for both repressive marks (*p* < 0.01). Similar analysis comparing only the larger nuclei (overlapping in *SUN/koi* and control groups) shows no significant difference between the fits, suggesting that mostly the smaller nuclei in the *SUN/koi* mutant contribute to the increased H3K27me3 and H3K9me3 repression. N = 5 larvae, *n* = 177 nuclei in control, N = 5 larvae, *n* = 295 nuclei in *SUN/koi.* For statistical significance * *p* < 0.05, ** *p* < 0.01. (**D**) Total DNA intensity plotted versus nuclear volume (log10 scale) for *SUN/koi* (red) and control (black). Linear mixed model analysis confirms significant difference between the *SUN/koi* and control fits (*p* < 0.01), with left-ward shift in the total DNA—nuclear volume relationship for the *SUN/koi* group. N = 5 larvae, *n* = 177 nuclei in control, N = 5 larvae, *n* = 295 nuclei in *SUN/koi*.

**Figure 4 cells-12-00932-f004:**
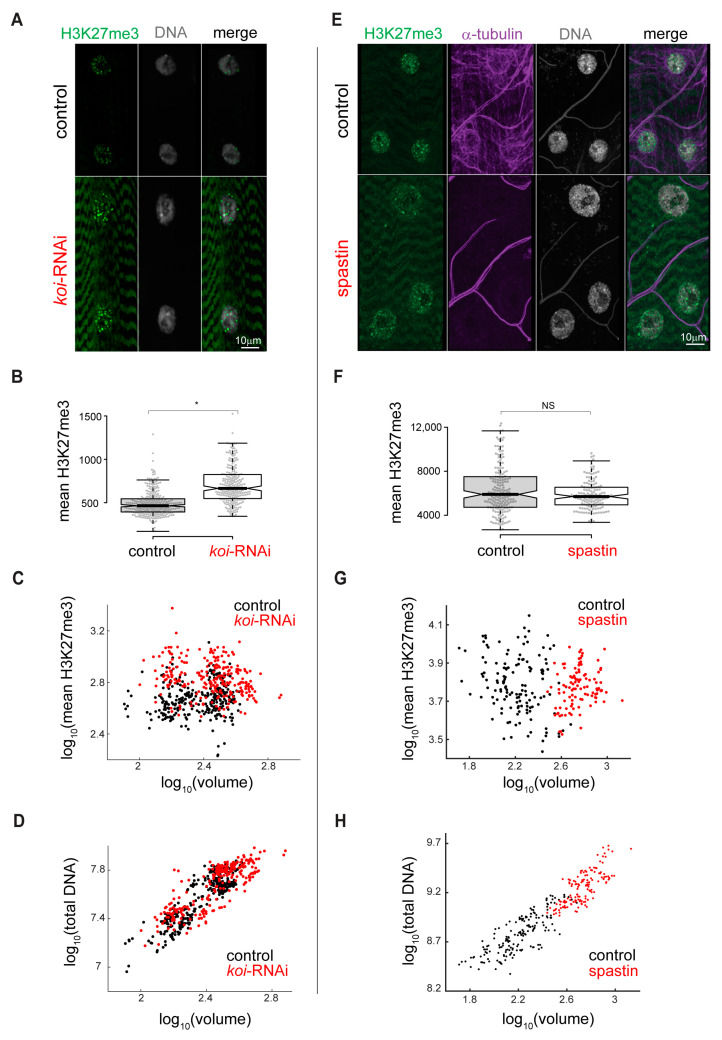
Increased H3K27me3 repression induced in temporal *SUN/koi* knockdown targeted to mature muscle fibers, but not in temporal disruption of the microtubule network. (**A**) Elevated H3K27me3 levels (green) and preserved DNA density (gray) in nuclei of mature muscle *SUN/koi*-RNAi. (**B**) Quantification of mean nuclear H3K27me3 intensity shows 45% increase in *SUN/koi*-RNAi group (*p* = 0.011, * statistically significant). (**C**) No correlation between mean nuclear H3K27me3 fluorescence intensity and the corresponding nuclear volume (log10 scale) in control (black) or *SUN/koi*-RNAi (red) groups (**D**) Preserved linear scaling between total DNA and nuclear volume (log10 scale) in *SUN/koi*-RNAi group. N = 5 larvae, *n* = 308 nuclei in control, N = 5 larvae, *n* = 250 nuclei in *SUN/koi*-RNAi. (**E**) Muscle nuclei of control and 1-day spastin overexpression demonstrate preserved H3K27me3 levels (green) upon MT network disruption (α-Tubulin label in purple), and preserved DNA density (gray), despite elevated nuclear volume. No change in mean H3K27me3 florescence intensity (NS=not statistically significant) (**F**) and its scaling with nuclear volume ((**G**), log10 scale) is maintained upon spastin overexpression. (**H**) Proportional increase in DNA intensity and nuclear volume (log10 scale) results in preserved global DNA density upon spastin over-expression. *n* = 5 larvae, *n* = 174 nuclei in control, N = 5 larvae, *n* = 140 nuclei in spastin.

**Figure 5 cells-12-00932-f005:**
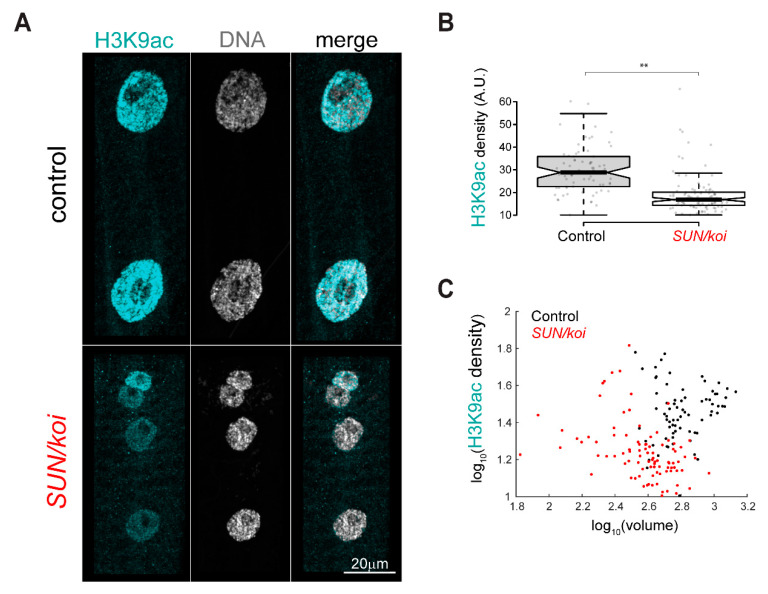
Reduced active H3K9ac chromatin density in *SUN/koi* mutated muscle fibers, independent of nuclear volume. (**A**) Muscle nuclei labeled with H3K9ac (cyan) and Hoechst for total DNA (gray). (**B**) Quantification of mean nuclear fluorescence intensity shows decreased H3K9ac in *SUN/koi* mutated muscle fibers. (**C**) Mean nuclear H3K9ac intensity is plotted against the corresponding nuclear volume (log10 scale) and shows no significant correlation with nuclear volume between the control and *SUN/koi* groups (left). N = 5 larvae, *n* = 73 in control, N = 4 larvae, *n* = 103 nuclei in *SUN/koi*. ** *p* < 0.01.

**Figure 6 cells-12-00932-f006:**
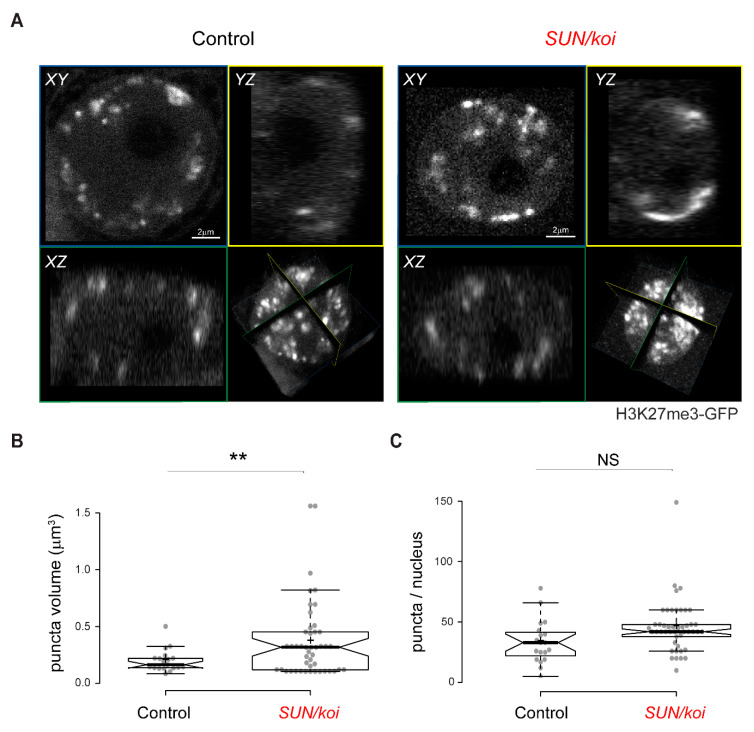
Live 3D imaging of H3K27me3 mintbody distribution indicates increased spread of the repressive mark in *SUN/koi* mutant muscle nuclei. (**A**) Middle confocal planes of control (**left**) and *SUN/koi* (**right**) muscle nuclei show punctuated repressive regions throughout the nucleus. Increased mean puncta volume (**B**) and unchanged number of puncta per nucleus (**C**) in the *SUN/koi* mutant group. NS =not significant. N = 4 larvae, *n* = 19 nuclei in control, N = 4 larvae, *n* = 46 nuclei in *SUN/koi* (** *p* < 0.01).

**Figure 7 cells-12-00932-f007:**
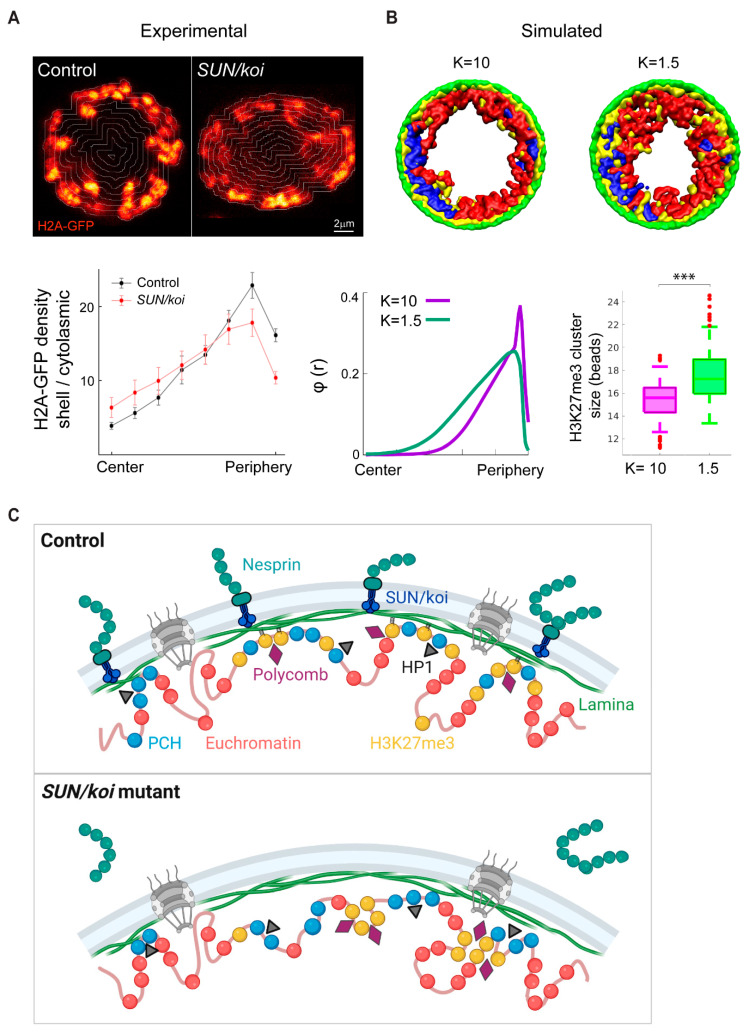
Experimental and computational evidence for chromatin redistribution towards the center as the mechanism for increased repressive clustering in the LINC mutant. (**A**) Radial chromatin distribution analysis from live, 3D imaging. Middle confocal planes of control (**left**) and *SUN/koi* (**right**) muscle nuclei labeled with H2A-GFP. Mean chromatin density is quantified for the radial shells and demonstrate decreased peripheral chromatin density with a central shift in the *SUN/koi* mutants (red, N = 4 larvae, *n* = 11 nuclei) compared to control (black, N = 3 larvae, *n* = 10 nuclei). (**B**) Computational model to describe the *Drosophila* chromosomes with a flexible bead-spring polymer chain containing three distinct regions: euchromatin (red), pericentromeric heterochromatin (blue), and H3K27me3 modification (yellow). The polymers are confined by a spherical wall comprising immobile lamin beads (green). The yellow (H3K27me3) beads are attracted to the green (laminar) beads and their bonding strength is modeled by a dynamic, harmonic (spring-like) potential K. Simulation snapshots (XY plane view), and quantification of the radial chromatin bead density demonstrate that for strong bonding of the yellow (H3K27me3) and green (laminar) beads, K = 10, most of the yellow clusters are located near the lamina, while for weak bonding, K = 1.5, the yellow clusters, diffuse within the spherical volume, resulting in a central chromatin shift. Box plots show increased H3K27me3 bead cluster size in the nucleoplasm with decreased bonding strength K (*p* < 0.001; *** statistically significant). (**C**) Schematic model to represent the mechanism for LINC mediated inhibition of chromatin repression. *SUN/koi* mutation decreases repressive chromatin binding to the lamina, resulting in increased H3K27me3 clustering in the nucleoplasm and redistribution of chromatin inward from the periphery (created with BioRender.com).

## Data Availability

The DamID data presented in this study are openly available in Mendeley Data at DOI: 10.17632/czjj5m7btr.1 https://data.mendeley.com/datasets/czjj5m7btr/1 (accessed on 17 March 2022).

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
