# Peer review of "The LINC Complex Inhibits Excessive Chromatin Repression"

_cells, 2023, doi:10.3390/cells12060932_

Round 1
Reviewer 1 Report
The authors analyzed the phenotype of SUN/koi mutant muscles in Drosophila 3rd instar larvae to investigate the contribution of the LINC complex to the epigenetic state. The authors compared the DNA binding profiles of Polycomb, HP1 and RNA pol II and showed that the binding profile between control and SUN/koi mutant muscle fibers was altered. They addressed the fact that the defective LINC complex is associated with increased chromatin repression, mainly through increased Polycomb binding, and decreased transcriptional activation. However, it is not clear whether this phenomenon is directly or indirectly induced by the SUN/koi mutation, since LINC complexes play a role in mechanotransduction to the nucleus and also function in the regulation of the cytoskeleton. I believe that the possibility that this phenomenon is caused by disruption of mechanotransduction or altered cytoskeletal control cannot be ruled out. In addition, although a number of papers have reported that disruption of the LINC complex affects the epigenetic state, I feel that the authors have not been properly cited. For example, Angelika Noegel's group reported that nesprin-2 knockout reduced NE-associated heterochromatin and HP1. They showed that nesprin-2 causes H3K9me3 accumulation in the center of the nucleus, whereas in control cells H3K9me3 is distributed throughout the nucleus and near the nuclear envelope (Rashmi et al., Nucleus 2012).
Other issues;
1. Figure 2A. To address the transcriptional repression in SUN/koi mutated muscle fibers, several representative genes should be analyzed.
2. RNA polymerase II can bind to DNA in the absence of transcriptional activation. It is therefore unlikely that the results of DamID with RNA polymerase II reflect the activated state of transcription. If the authors want to determine the activation state of transcription, it is better to use phosphorylation-specific RNA polymerase II. https://pubmed.ncbi.nlm.nih.gov/34854870/Figure 2C includes a list of specific genes of interest. If specific genes are described, the other data should be included in the supplemental data.
3. Figure 2C contains a list of specific genes of interest. If specific genes are described, the other data should be included in the supplementary data. In addition, Blank columns in tables need to be filled in.
4. Figure 2D shows upregulated HP1, Pol II and Polycom. For RNA polymerase II, I think it would be easier to analyse if down-regulated versions were included.
5. “Taken together our genome-wide analysis indicates that defective LINC complex as-sociates with increased chromatin repression,” (p6) Why is it underlined?
6. The authors stated that “Figure 3B shows quantification for the mean nuclear fluorescence intensity with 43% increase in H3K27me3 (p<0.05), and 82% increase in H3K9me3 (p<0.01). The overall increased inten-sity of repressive chromatin marks is in agreement with the increased Polycomb, and HP1a binding observed with the targeted DamID” (p8). However, in the case of DamID, the Polycomb signal was significantly increased in SUN/koi, but to a lesser extent in HP1. In contrast, the signal of H3K9me3 is significantly increased in fluorescent antibodies. In other words, it could be argued that there is not much correlation between DamID and fluorescent antibodies.
7. The linear mixed model analysis and the fitted mixed model analysis require detailed explanation.
8. It would be better to include Polycome and HP1 in Figure 7C.
Reviewer 2 Report
My first impression reading the manuscript is very positive.I think it requires just some rearrangement of the text (too much discussion already in the results) and improved figure captions.
The abstract will require some introduction into the topic.
A small scientific point regards the quantification of the H3K27me3 chromatin density (e.g. Fig. 3) because the images have a high background....
Reviewer 3 Report
In this manuscript, Pavlov and his colleagues analyzed the genome-wide binding profile of Polycomb, HP1A, and RNA polymerase â…¡ (Pol â…¡) upon loss of nucleoskeleton and cytoskeleton (LINC) complex (SUN/koi mutant) in muscle fibers of Drosophila. They found that the chromatin tends to be more repressed epigenetically (increased Polycomb and HP1A, decreased Pol â…¡). Furthermore, by analyzing imaging data, the authors showed that the two repressive histone markers, H3K9me3 and H3K27me3, are increased; meanwhile, the histone activation marker H3K9ac is decreased. Interestingly, the authors found that loss of LINK complex causes repressive chromatin unbound from lamin, which leads to increased repressive H2K27me3 clustering in the nucleo-plasm. Overall, the authors’ findings are very attractive and essential. However, I believe some explanations and experiments need to be done to better support the authors’ conclusions.
Comments:
1. In figure 1B, the authors analyzed three genes as examples to show the Polycomb, HP1A, and Pol â…¡ changes in SUN/koi mutant. However, these changes seem only partially within a gene region. It raises the question of whether these changes can lead to transcriptional changes of the genes. Could authors do RNAseq in SUN/koi mutant fly? Or at least do RT-qPCR for these genes in SUN/koi mutant fly to check the mRNA levels.
2. The authors showed the H3K27me3 increased upon SUN/koi mutant by analyzing the image data, the results are interesting and promising. To follow up on these results, it is worth checking the H3K27me3 on the gene level by performing ChIPseq experiments, and then the authors should have more details on how H3K27me3 clusters alter upon SUN/koi mutant. Also, the RNAseq experiment might be a good idea to test the transcriptional changes of genes at the genome-wide level. This will further support the authors’ conclusions that epigenetic alteration caused by SUN/koi mutant can change the transcription of genes.
